# Coffee By-Products and Their Suitability for Developing Active Food Packaging Materials

**DOI:** 10.3390/foods10030683

**Published:** 2021-03-23

**Authors:** Gonçalo Oliveira, Cláudia P. Passos, Paula Ferreira, Manuel A. Coimbra, Idalina Gonçalves

**Affiliations:** 1CICECO–Aveiro Institute of Materials, Department of Materials and Ceramic Engineering, University of Aveiro, 3810-193 Aveiro, Portugal; gvso@ua.pt (G.O.); pcferreira@ua.pt (P.F.); 2LAQV-REQUIMTE, Department of Chemistry, University of Aveiro, 3810-193 Aveiro, Portugal; cpassos@ua.pt (C.P.P.); mac@ua.pt (M.A.C.)

**Keywords:** polysaccharides, phenolics, lipids, circular economy, composites, bioplastics

## Abstract

The coffee industry generates a wide variety of by-products derived from green coffee processing (pulp, mucilage, parchment, and husk) and roasting (silverskin and spent coffee grounds). All these fractions are simply discarded, despite their high potential value. Given their polysaccharide-rich composition, along with a significant number of other active biomolecules, coffee by-products are being considered for use in the production of plastics, in line with the notion of the circular economy. This review highlights the chemical composition of coffee by-products and their fractionation, evaluating their potential for use either as polymeric matrices or additives for developing plastic materials. Coffee by-product-derived molecules can confer antioxidant and antimicrobial activities upon plastic materials, as well as surface hydrophobicity, gas impermeability, and increased mechanical resistance, suitable for the development of active food packaging. Overall, this review aims to identify sustainable and eco-friendly strategies for valorizing coffee by-products while offering suitable raw materials for biodegradable plastic formulations, emphasizing their application in the food packaging sector.

## 1. Introduction

Since the 1960s, world plastic production has increased, reaching 360 million tons in 2018 [1]. A great part of the produced plastics are single-use materials made of nonbiodegradable, petroleum-based molecules, with the most part being landfilled or discarded in water streams, ending in oceans as microparticles. Seeking to diminish the global nonbiodegradable plastic environmental impact, efforts are being made to use recyclable, biodegradable, or compostable plastic packaging materials by 2030 [2]. In this context, bioplastics have attracted the attention of the plastics industry. Bioplastic is a plastic material which is either bio-based or biodegradable [3], whereby none of the compounds released from the plastic formulations present environmental toxicity. The synthesized polyester polylactic acid (PLA) is one of the most commonly used polymers for bioplastic production [4]. However, when compared with the most widely used petrochemical plastics, PLA has low thermal stability [5] and poor water vapor and gas barrier properties [6], limiting its application range. PLA has been obtained from the fermentation of different starch-rich vegetables, such as corn, beet, or wheat bran [7], representing a societal disadvantage, since these raw materials are directly used in human and animal nutrition. To avoid any kind of competition between different industry sectors, biomolecules of interest for developing bioplastics must be recovered from agri-food by-products. This goal has been pursued by using agri-food by-products such as apple pulp waste as a carbon source for microbial fermentation in the production of bioplastic polymers, namely, polyhydroxyalkanoates (PHAs) [8]. Another strategy is the use of starch and lipidic molecules recovered from potato and rice by-products in the development of starch-based films [9,10,11,12]. The film-forming ability of starch allows its use as the main polymer matrix for the development of bioplastic materials, while lipids are used to confer hydrophobicity and plasticity/elasticity. Herein, instead of simply being discarded, agri-food by-products may be introduced into new bioplastic processing chains, promoting a circular economy between the agri-food and plastic sectors. Furthermore, agri-food by-products can be used to confer biodegradability upon petroleum-based plastics, or to increase the biodegradation rate of synthetic plastics, as observed when potato peel waste is incorporated into polyethylene and polypropylene formulations [13], and when cellulose fibers obtained from sugar cane leaf are added into polyvinyl alcohol (PVA) formulations [14].

Coffee production generates a number of by-products derived from the plant, namely, the flowers, which are available at pre-harvesting, by-products derived from the harvesting of the cherries, and the leaves, stems, twigs, and wood, mainly left post-harvesting, which have potential uses in the food sector [15]. The coffee industry, while processing the cherries, also generates a variety of by-products with distinct chemical compositions, including coffee pulp (CP), mucilage (CM), parchment (CPm), husks (CH), silverskin (CS) and spent coffee grounds (SCG). These by-products have been proposed for multiple applications due to their high biodegradability, namely, as substrates for the production of enzymes, bioethanol, and biogas by microorganisms [16]. Coffee by-products have also been suggested as biosorbents for water treatment [17,18,19] or as a source of dietary fiber and food ingredient active compounds, including chlorogenic acids and caffeine [15,20,21]. The diversity of coffee by-products represents a source of different molecules suitable for the development of plastics with different physicochemical and/or biological properties. This review focuses on the feasibility of using coffee by-products for the development of plastics with improved physicochemical, mechanical, barrier, and biodegradability properties, detailing their chemical composition and discussing their potential for use either as polymeric matrices or as a functional additive for food packaging.

## 2. Coffee By-Products and Their Potential for Use in the Development of Plastics

### 2.1. The Coffee Industry and Its By-Products

The coffee industry is one of the largest food industries in the world. It is divided into two main sectors: the first includes the separation of shell and mucilaginous part from the coffee cherries in order to recover coffee beans, implemented in coffee-producing countries; the second is responsible for the roasting and brewing transformation steps, which occur after shipping the beans to coffee distributing and consuming countries. According to the International Coffee Organization (ICO), global coffee production reached 169 million 60 kg bags in 2019/2020 [22]. The great demand for this product has led to the production of an excessive amount of by-products during all coffee processing steps (Figure 1).

Coffee flowers are known as a source of caffeine and trigonelline, as well as phenolic compounds [23]. However, as no toxicological data seems to be available for their use in foods [15], their potential for valorization is still limited. Coffee leaves contain a large diversity of compounds [24] which may be valorized as bioactive compounds with antioxidant, anti-inflammatory, antihypertensive, antibacterial, and antifungal activities [15]. Although no studies exist for their application in food packaging, coffee leaves seem to be a very promising source of compounds for this purpose.

Coffee cherry can be processed by two different methods, designated as dry and wet processing methods [25], which yield different by-products. CH result from the dry processing method. It correspond to the outer layers (from endocarp to epicarp) removed from the dried cherry, representing 45% of fresh coffee cherry weight [26]. This means that per 100 kg of dry processed fresh coffee cherries, 45 kg of CH are obtained. CH are also named “cascara”, corresponding to a mixture of skin, pulp, mucilage, parchment, and part of silverskin resulting from the dry processing method [27]. “Sticky” CH is an additional by-product that can be obtained when the parchment layer is not removed with the outer layers, resulting in a by-product with a high level of protein and low level of fiber [28,29]. In the wet processing method, CP, also named “fresh cascara”, is a by-product that corresponds to a mixture of coffee cherry outer skin and pulp layer that can be separated by depulping in water [27]. CM is a sticky mucilage layer obtained by the mechanical action of the applied equipment or by the action of fermenting enzymes when the depulped coffee beans are placed in fermentation tanks. After washing, drying, and dehulling the fermented coffee beans, CPm is the last by-product of this process. From 100 kg of wet processed fresh coffee cherries, 39 kg of CP, 22 kg of CM, and 39 kg of CPm are obtained [27], which means that part of the water added during wet processing is retained in CP and CM. All these by-products remain in coffee-producing countries, being incinerated or used for biofuel production [30,31], limiting their reuse in other applications.

During green coffee roasting, bean blow-up leads to the release of a thin layer called CS. When the roasted coffee beans are ground and used for brew preparation, the compounds that are not extracted by hot water are named SCG. For 100 kg of green coffee beans, around 2.1 kg of CS [27] and 65 kg of SCG [16] are produced. Since 100 kg of fresh coffee cherries give rise to around 21 kg of green coffee beans [32], 0.4 kg of CS and 14 kg of SCG are obtained from 100 kg of coffee cherries. Although most of the CS and SCG is usually incinerated or landfilled, these materials have the potential to be used as food ingredients [33,34], mainly due to their dietary fiber-rich composition. Moreover, the use of SCG has been proposed for diverse applications, such as cosmetics, animal feed, bioethanol production, adsorbents, and fertilizers [27].

The valorization of coffee by-products and potential applications depend on their chemical composition (Table 1).

#### 2.1.1. Coffee Pulp and Mucilage

Coffee pulp (CP) is one of the main coffee wet processing by-products. CP has a high moisture content (78–81% wt) [31,35], due to the incorporation of water during the washing of coffee cherries before the depulping process. This high-water activity promotes microbial spoilage, a problem faced by all by-products where the water content is not decreased to a level that promotes stabilization. On a dry weight basis, CP is mostly constituted of cellulose (36%), pectic polysaccharides (21%), a fraction of alkaline soluble polysaccharides defined as hemicelluloses (9%), and free sugars (5%) [36]. CP pectic polysaccharides comprise 80% galacturonic acid, 63% methyl esterification degree (DE), 6% acetylation degree (DA), and high molecular weight (4 × 10^5^ g/mol), giving rise to a gel-forming ability in the presence of high sucrose concentration and low pH [37]. Nevertheless, the CP content in pectic polysaccharides and free sugars is lower when compared with other vegetable-derived pulps such as apple pomace [38]. This may be a consequence of carbohydrate degradation caused by the action of endogenous enzymes, since wet CP is not immediately dried and/or frozen after the depulping process. CP is also made up of proteins (9%), alkaloids (1%), lipids (0.8%), and phenolic compounds (3 mg g^−1^ of gallic acid equivalents, GAE) [36]. The CP protein content is often estimated according to the Kjeldahl method and using the N × 6.25 conversion factor, after the total nitrogen determination. However, CP is also composed of other nitrogenous compounds, namely caffeine (C_8_H_10_N_4_O_2_) and trigonelline (C_7_H_7_NO_2_). Therefore, most reported CP protein content values may be overestimated. Concerning CP lipids, they are derived from the cherry skin (epicarp), with cutin being the most abundant compound. Cutin is a polyester formed by esterified ω-hydroxy and ω-hydroxy-epoxy fatty acids and glycerol [39]. The CP phenolic composition includes hydroxycinnamic acids (59%), flavanols (17%), and hydroxycoumarins (6%) [40].

Coffee mucilage (CM) has a high moisture content (84% wt) [41], in the same order of magnitude as that of CP. On a dry weight basis, CM is mainly constituted by pectic polysaccharides (30%), hemicelluloses (18%), proteins (17%), and cellulose (8%) [48]. CM pectic polysaccharides have 52% galacturonic acid, 85% DE, 6% DA, and gel-forming ability [52]. However, this ability may be compromised, given the lower molecular weight of CM pectic polysaccharides (1.2 × 10^4^ g/mol) when compared with CP [48]. Also, the higher content of pectic polysaccharides observed in CM contradicts a previous study, where it was reported that CP had a pectic polysaccharide content two times higher than CM [59]. The presence of pectic polysaccharides in the chemical compositions of CM and CH enhances their potential to be directly applied in the development of bioplastics. CM pectic polysaccharides derived from fermentation and from the mechanical removal process have similar chemical compositions. Fermentation only induces a slight decrease of the pectic polysaccharide intrinsic viscosity and average molecular weight, and a two-fold increase of its DA [60], which would not compromise the valorization of fermentation-derived CM.

#### 2.1.2. Coffee Husks

Coffee husks (CH) are the only by-product derived from coffee cherry drying and dehusking (Figure 1). CH have a moisture content of 13–15% [29,43], depending on the drying process time. On a dry weight basis, CH are constituted of lignin (38%), cellulose (28%), and a fraction of hemicelluloses (25%) rich in xylose residues [50], possibly derived from glucuronoxylans, a polysaccharide usually present in lignified tissues [61]. CH are also made up of proteins (8–11%), lipids (1–3%), and caffeine (1%) [29,53]. CH ashes account for 3–7% [29,55]. As in CP, the protein content may be overestimated due to the presence of other nitrogenous compounds. The lipid fraction may derive from the cherry skin cutin and also from silverskin partially removed during the dehusking process [27]. CH are rich in phenolic compounds (13 mg g^−1^ of GAE), mainly caffeic and chlorogenic acids [58]. The paucity of data available shows that for the valorization of CH, much more research is required.

#### 2.1.3. Coffee Parchment and Silverskin

Coffee parchment (CPm) is a fibrous endocarp that covers the coffee cherry epidermis and endosperm. Since this by-product is obtained after drying and dehulling the beans, its moisture content is low (9% wt) [42]. On a dry weight basis, CPm is composed of xylans (35%), lignin (32%), and cellulose (12%) [49]. CPm ashes account for 1% [55]. This composition shows the insoluble nature of CPm, with possible application in the development of food packaging plastics. CPm has a high water (3 mLg^−1^) and oil (4 mLg^−1^) holding capacity [49], allowing its use as barrier, avoiding condensation of water inside food packaging, as well as the migration of fat from greasy foods. CPm is also composed of caffeine (0.13%) and phenolic compounds (2 mg g^−1^ of GAE), namely gallic acid, chlorogenic acids, *p*-coumaric acid, and sinapic acid [42], which provide antioxidant activity. As observed for CH, the potential for the of valorization of CPm is significant.

Coffee silverskin (CS) is a thin tegument of the coffee bean outer layer, being the most abundant by-product associated with coffee roasting. It has a low moisture content (4–7% wt) [34,44,45,46], facilitating its storage and direct use. CS composition is similar to that of CPm, given their proximity inside the cherry. On a dry weight basis, CS is constituted of polysaccharides (40%), mainly cellulose (59%), with a small proportion of xylose (19%), arabinose (9%), galactose (9%), and mannose (4%) [51]. No information about CS polysaccharides glycosidic-linkage composition has been yet reported. CS also contains lignin (29%) [51], proteins (19%), and lipids (2–5%), while CS ashes account for 5–7% [34,44,45,51]. Furthermore, CS contains caffeine (1%) and phenolic compounds (2% *w*/*w* GAE), mainly chlorogenic acids as 3-*O*-caffeoylquinic acid and 4-*O*-caffeoylquinic acid [57]. As in CP and CH, the protein content of CS may be overestimated because of other nitrogenous fractions present in CS, since most of the reported studies used the nitrogen content for protein quantification in non-purified fractions. The presence of lipids in CS, in contrast to CPm, is due to its proximity to the cherry endosperm (coffee bean), which has a significant fraction of lipids (8–18% of the green coffee bean dry weight) [41]. Because CS derives from the roasting process, it is also composed of melanoidins (5%), which are nitrogenous high molecular weight heterogeneous polymers formed through Maillard reactions during roasting [34].

#### 2.1.4. Spent Coffee Grounds

Spent coffee grounds (SCG) are wet solid residues (61% of moisture [47]) that remain after coffee brewing, being produced all over the world where coffee is consumed. The high accessibility of this coffee by-product facilitates the study of its chemical characterization and further applications. On a dry weight basis, SCG are constituted by polysaccharides (66%), mainly galactomannans (50%), arabinogalactans (25%), and cellulose (25%) [47,62]. Coffee galactomannans are high molecular weight polysaccharides with low branching degree, arranged by a backbone of (β1→4)-linked mannose residues, with O-6 single (α1→6)-linked galactose and single (1→5)-linked arabinose residues [63]. They are water-soluble and form highly viscous and stable aqueous solutions with film-forming ability [64,65], which makes them suitable raw materials for the production of edible and biodegradable films or coatings for food applications [66]. The amount of melanoidins in SCG is estimated to be 16%, with 5% of proteins [47], 13–15% lipids [47,54], 0.01–0.5% of caffeine, and phenolic compounds (1–2% *w*/*w* GAE) [56]. SCG phenolic composition includes mainly chlorogenic acids (85%), such as 3-*O*-caffeoylquinic acid, 4-*O*-caffeoylquinic acid, and 5-*O*-caffeoylquinic acid, and also caffeic acid (6%) [67]. Ashes account for 2% [56]. SCG-derived oil is composed mainly of linoleic (45%) and palmitic (38%) acids [68], and also by diterpenes (15%), namely kahweol, cafestol and 16-O-methylcafestol [69]. Although slight variations can occur in the chemical composition of SCG according to the coffee brew extraction conditions [70] and the composition of the roasted coffee beans, which depends on the coffee species and postharvest processing conditions, the SCG overall composition seems to be highly consistent for food packaging applications.

### 2.2. Coffee By-Products for the Production of Plastics 

The use of coffee by-products in the development of sustainable plastic formulations for food packaging follows two main strategies: (1) the use of crude coffee by-products as functional additives for plastics; or (2) the use of coffee by-product-derived extracts with film-forming ability or functional properties (Figure 2).

#### 2.2.1. Crude Coffee By-Products as Functional Additives for Plastics

The crude form of coffee by-products, namely CH, CS, and SCG, has been incorporated into plastic formulations. This strategy presents a zero-waste approach, since all the crude fractions can be used without generating residues. Many studies have reported the incorporation of crude coffee by-products into nonbiodegradable plastics (Table 2), with the goal of conferring biodegradability upon petroleum-based materials. This can be seen as a first step towards a full bio-based and biodegradable formulation which is able to provide the same functionalities as petroleum-based materials.

##### Nonbiodegradable Formulations

CH [71] and SCG [78] powder can be incorporated into polypropylene (PP)-based formulations, and CS powder [77] into high-density polyethylene (HDPE)-based formulations. These composites showed increased rigidity and decreased elongation at break, due to the poor interfacial adhesion between coffee by-products and the polymeric matrices. This incompatibility can be caused by the different nature of molecules in the same plastic formulation. While coffee by-products are mainly constituted of carbohydrates rich in hydroxyl groups, thus possessing a polar nature, the matrix of petroleum-based polymers is constituted by nonpolar hydrocarbons.

One strategy to increase the interfacial adhesion between coffee by-products and petroleum-based polymers is the addition of malleated compatibilizers into the formulation. For instance, maleic anhydride grafted PP [72] and PE [73] were added, together with CH powder, to PP and PE-based formulations, respectively. The maleic anhydride groups interacted with the hydroxyl groups of CH carbohydrates through covalent bonds, while the long molecular chains of malleated compounds entangled the hydrophobic petroleum-based matrix, increasing the constituent compatibility and giving rise to with good interfacial adhesion [83]. Also, the incorporation of CH powder together with maleic anhydride grafted PP in a PP-based formulation reduces the flammability of the composite [74] and the carbon footprint [75], compared with neat PP composites.

Another strategy to increase compatibilization is the chemical modification of crude coffee by-products before their addition into petroleum-based formulations. Herein, small modification techniques, such as alkaline treatment or esterification of coffee by-products are proposed. The performance of an alkaline treatment upon crude CH seems to be effective at improving the mechanical and thermal performance of PP-based composites [76]. Alkali treatment removes alkaline soluble polysaccharides, lipids, impurities, and a fraction of the lignin from coffee by-products, exposing more cellulose molecules and increasing the number of reaction sites [84]. For instance, alkali treated and bleached SCG, and its mixture with PP-based formulations, together with silane and styrene-ethylene-butene-styrene-graft-maleic anhydride as coupling agents, showed improved composite interfacial adhesion and mechanical properties by establishing stronger interactions with the polymeric matrix [81]. Alternatively, the addition of SCG powder esterified with palmitoyl chloride can originate better particle dispersion and a decrease in water uptake upon PP-based composites [80]. During esterification, the polar carbohydrate hydroxyl groups of coffee by-products react with acetyl groups, decreasing the molecule polarity and increasing their compatibility with the nonpolar hydrocarbon-based matrix [85]. This chemical modification also leads to the hydrophobization of coffee by-product compounds. In a combination of the two chemical modification techniques previously described, the incorporation of alkali treated and esterified CS powder into high-density polyethylene (HDPE)-based formulations can be performed to develop composites with decreased water absorption [77]. All composites containing CS are brownish, the intensity of which increased with CS concentration, due to the presence of melanoidins [34]. Another strategy that increases the compatibilization between SCG and a PP-based matrix is the removal of the SCG lipid fraction [79]. The defatting process leads to a better dispersion of SCG into a malleated-PP-based matrix, improving its interfacial adhesion and producing composites with better water resistance. SCG acid hydrolysis can remove the majority of carbohydrates, increasing the accessibility to SCG phenolic compounds. When added to a polyethylene (PE)-based formulation, acid hydrolysis treated SCG are able to increase the antioxidant ability and biocompatibility of the resultant films, enhancing their potential for use for the preservation of food lipids [82].

Although a large number of studies are already available in the literature concerning the addition of crude coffee by-products to increase the biodegradability of petroleum-based plastics, no biodegradability studies have been described. There is a need for biodegradation tests in order to evaluate if crude coffee by-products have the ability to confer biodegradability upon petroleum-based materials, minimizing their negative impact on the environment.

##### Biodegradable Formulations

Crude coffee by-products have also been used as additives of biodegradable plastic formulations (synthetic and bio-based) (Table 3).

Concerning the direct use of coffee by-products in their crude form, CH powder, when added to polycaprolactone (PCL)-based formulations, increases their biodegradation rate by acting as a support for microorganism adhesion [86]. CH and CPm can also be used as reinforcing fillers of polyhydroxybutyrate (PHB)-based compounds, a biodegradable thermoplastic polyester produced by bacterial fermentation, increasing their water absorption and thermal stability by delaying their degradation temperature [89]. Moreover, crude CS can be used as an additive of PBAT and poly(3-hydroxybutyrate-*co*-3-hydroxyvalerate) (P(3HB-*co*-3HV)) blended formulations, leading to the development of composites with increased rigidity and antioxidant properties [90], in accordance with the reported antioxidant activity verified in CS extracts [57,101,102]. Crude CS has the potential to integrate bio-based industrial injection molding formulations for coffee capsules, as demonstrated by its addition to P(3HB-*co*-3HV), together with acetyl tributyl citrate and calcium carbonate as a plasticizer and inorganic filler, respectively. However, the low interfacial adhesion between CS and P(3HB-*co*-3HV) matrix obtained after injection molding of the composites [92] needs to be improved.

To improve the compatibility between coffee by-products and synthetic biodegradable polyesters, the modification of coffee by-products by silane-based compounds has been proposed. Silane molecules, such as (3-glycidoxypropyl) trimethoxysilane, have bifunctional groups that can act as coupling agents between the hydroxyl groups of polysaccharides and the epoxy of the nonpolar polyester that constitute the synthetic matrix [103]. It was claimed that the incorporation of silane treated crude CH into polybutylene adipate terephthalate (PBAT)-based formulations can yield composites with higher hydrophobicity and stiffness than neat PBAT-based composites, as well as lower production costs (32%), making these materials competitive with conventional commercial polymers [87]. Better interfacial adhesion between CS and PBAT/P(3HB-*co*-3HV) blend can also be obtained after performing silane treatment on crude CS [91]. Tetraethyl orthosilicate can also be used to increase the homogeneity, water resistance, and biodegradability of maleic-anhydride-grafted PLA/SCG-based formulations [94].

The torrefaction of coffee by-products before their addition into bioplastic-based formulations is another strategy adopted to increase the biocompatibility between crude coffee by-products and polyesters. In this context, torrefaction has been used as a strategy to produce coal fuel from biomass at 200–300 °C under inert nitrogen gas atmosphere [104], making it possible to increase the hydrophobicity of the material by the dehydration of cellulose and lignin, thereby reducing the number of hydroxyl groups available [105]. The addition of torrefied CH into PLA-based formulations increases the mechanical resistance and improves the thermal stability by delaying the degradation time of the corresponding mold injected specimens [88]. Moreover, the torrefaction of SCG increases the hydrophobicity of PBAT/SCG-based composites, enhancing the potential of this treatment to develop hydrophobic food packaging materials, which can extend food shelf-life by preventing interaction with water [95].

When applied to polar polymeric matrices, coffee by-products can be blended with other polar compounds, such as polysaccharides, to develop homogeneous materials, as observed when chitosan was incorporated into PVA/SCG-based formulations [96]. As a result, SCG can be successfully incorporated into the polymeric matrix, yielding composites which are suitable for adsorbing pharmaceutical contaminants in water [96]. This property may also be relevant for active food packaging.

When coffee by-products are added into polysaccharide-based formulations, good compatibilization can be achieved without previous treatment or incorporation of any compatibilizer agent into the plastic formulation. The incorporation of crude SCG powder into corn starch-based formulations leads to the development of films with increased tensile strength [97]. Similarly, the incorporation of crude SCG powder into cellulose-based formulations leads to the development of films with decreased light transmission (high light resistance), with potential for use in vegetable packaging [98]. Moreover, crude SCG has the potential to increase the water tolerance of pectin-based films [99,100]. Regarding crude CS, its addition into potato starch-based formulations led to the development of films with increased elasticity, stretchability, and water resistance while conferring antioxidant and UV-protective abilities upon the pristine potato starch-based films [93]. Therefore, blending coffee by-products with polysaccharide-based formulations can lead to the creation of materials which are competitive with nonbiodegradable food packaging plastics.

#### 2.2.2. Coffee By-Product-Derived Extracts with Film-Forming Ability or Functional Properties

Coffee by-products are a source of compounds with film-forming ability or functional molecules, such as lipids, phenolics, and polysaccharides, which are suitable for the production of plastics with improved performance for food packaging (Table 4). 

Regarding lipid-rich extracts, among all coffee by-products, only SCG have been used due to their high lipid content (13–15% dry wt basis) (Table 1). SCG-derived oil can be incorporated into PLA-based formulations, giving rise to composites with increased toughness and suitable for 3D-printing applications, due to the uniform distribution of SCG-oil molecules within the polymeric matrix [106]. Also, the addition of SCG fatty acids-rich extracts combined with diatomite leads to the development of multifunctional PLA-based films with increased interfacial adhesion and decreased oxygen permeability [107]. Herein, the molecules of SCG-extract and diatomite act as reinforcing fillers, hindering the diffusion of air molecules through the PLA-based matrix.

Phenolic-rich extracts obtained from coffee by-products have been used to confer active properties upon plastic polysaccharide-based formulations. Phenolic-rich extracts obtained from CP develop yellowish films with increased water resistance (decreased water vapor permeability and water solubility), antioxidant, and antimicrobial properties when incorporated into chitosan-based formulations [108], having the potential to prevent food oxidation reactions when used as packaging due to the radical scavenging activity of coffee phenolic compounds [109]. Moreover, the inherent antimicrobial activity of coffee phenolic compounds, namely chlorogenic acids, can also confer protection against microbial spoilage [110]. Similarly to chitosan-based formulations, corn starch-based films with increased tensile strength, decreased water vapor and oxygen permeabilities (by 30% and 50–85%, respectively), and antioxidant/antibacterial activities have been reported by the addition of CH hydrothermal aqueous extracts [58]. These extracts have also been used to confer antioxidant activity and to decrease the oxygen permeability of corn starch/PLA-based films [111]. Gellan gum-based films with antifungal properties have also been prepared with phenolic-rich extracts obtained from CPm [42]. Phenolic-rich extracts recovered from SCG can also originate from PVA/cassava starch- [112] and cassava starch- [113] based films with active properties, namely antioxidant, antimicrobial, and antibacterial activities. Although the increased tensile strength of the films has been attributed to the phenolic compounds present in the extracts, it is possible that this effect is due to co-extracted polysaccharides. In addition, the co-extraction of alkaloids, such as caffeine, may also contribute to the antimicrobial properties of the films [114]. 

Regarding polysaccharides, cellulose-rich materials have been used to enhance the physicochemical and mechanical performance of bioplastics. After delignification and bleaching CH, the obtained cellulose fibers (10–50 μm diameter and 1–3 mm length [115]) are capable of increasing the toughness of corn starch-based formulations [58]. After hydrolysis of CH-derived cellulose, the resulting cellulose nanocrystals (2–20 nm diameter and 100–600 nm length [116]) are capable of increasing the traction resistance of corn starch/PLA-based matrices [111]. On the other hand, cellulose nanocrystals-derived from CS (8 nm diameter and 80 nm length) decrease the water vapor and oxygen permeabilities of PLA-based matrices [117]. Moreover, galactomannan and arabinogalactan-rich extracts derived from SCG increase the light barrier, tensile resistance, and surface hydrophobicity of carboxymethyl cellulose-based films [118].

Coffee by-product-derived polysaccharides with film-forming ability can be directly used to form bioplastics. Pectic polysaccharides (DE 85%) obtained from CM can be used as polymeric matrices in bioplastics production, yielding biodegradable films with rigidity and water insolubility [52] for use as food packaging. Moreover, polysaccharide-rich extracts obtained from SCG can yield light-brownish films [64], with potential to protect foodstuffs from light when used as packaging. However, these SCG-derived films are heterogeneous and possess surface aggregates, which compromise their mechanical performance. While the brownish coloration derives from SCG melanoidins, the presence of aggregates (visible dark brown spots) may be related with the formation of complexes between polysaccharides and chlorogenic acids during film development. The partial removal of (β1→4)-linked glucose residues from this extract (12%) by enzymatic hydrolysis can be performed to obtain a fraction rich in galactomannans which is able to form light-brown heterogeneous films with higher rigidity. The presence of less (β1→4)-linked residues in the extract increases the galactomannan intermolecular bonds [65]. Future research in this area is essential to extend the application range of these materials, e.g., making them suitable for food packaging applications.

## 3. Conclusions and Future Perspectives

This review addresses the potential of using coffee by-products either as crude additives or as coffee by-products-derived extracts rich in lipids, phenolics, and polysaccharides, to improve the physicochemical, mechanical, barrier, and biodegradability properties of film-forming materials, thereby contributing to food packaging sustainability. Moreover, coffee by-products can be used as a source of compounds with film-forming ability, such as pectic polysaccharides from pulp and mucilage, and galactomannans from spent coffee grounds, in the development of bioplastics. Although coffee husks, coffee silverskin, and spent coffee grounds are the most studied plant parts regarding food packaging, all coffee by-products have potential for this purpose. Due to their inherent physicochemical constitution, coffee by-products may give rise to packaging materials with decisive properties for food preservation, such antioxidant activity, antimicrobial properties, increased mechanical resistance, and surface hydrophobicity, as well as improved gas barrier performance. These properties enable coffee by-products to extend their application range to the active food packaging sector while contributing to the circular economy. To this end, the dehydration of coffee byproducts, or any other procedure which allows them to retain their quality, is a requirement. Most studies to date were only performed on a laboratory scale, using solvent casting technology, and failed to provide information about the biodegradability performance of the developed materials. Aiming to fulfil the requirements of the active food packaging industry, the evaluation of the biodegradability, processability, and upscaling potential of coffee by-product-based materials is a significant challenge.

## Figures and Tables

**Figure 1 foods-10-00683-f001:**
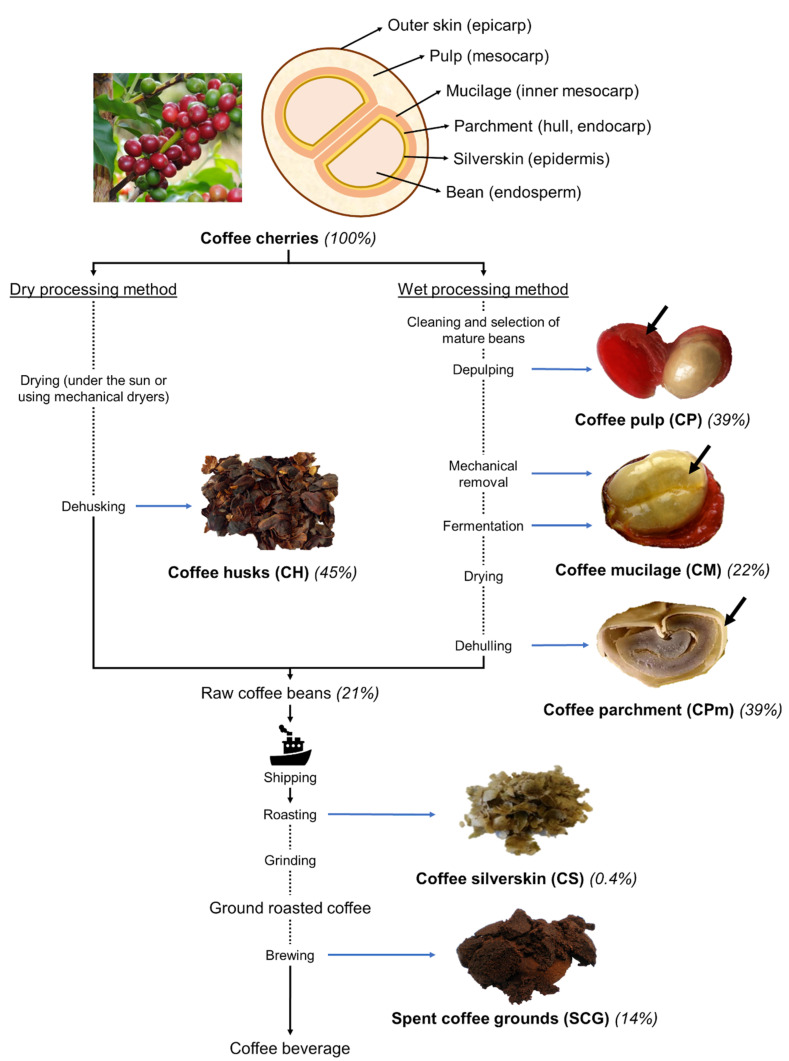
Schematic representation of coffee cherry structure and coffee processing-derived by-products. The percentages refer to the amount of each by-product obtained from fresh coffee cherries.

**Figure 2 foods-10-00683-f002:**
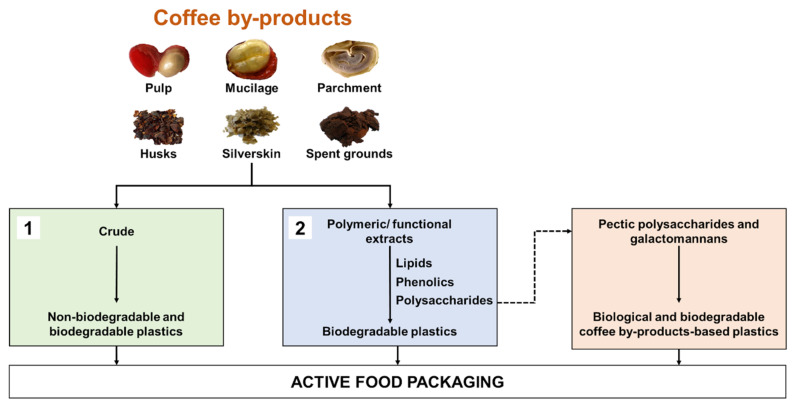
Strategies for reusing coffee by-products in the development of plastic-based formulations: (1) the use of crude coffee by-products as functional additives for plastics; or (2) the use of coffee by-products-derived extracts with film-forming ability or functional properties.

**Table 1 foods-10-00683-t001:** Chemical composition of coffee by-products.

Composition	Pulp (CP)		Mucilage (CM)		Parchment (CPm)		Husks (CH)		Silverskin (CS)		Spent Coffee Grounds (SCG)	
Moisture (% wt)	78–81	[31,35]	84	[41]	9	[42]	13–15	[29,43]	4–7	[34,44,45,46]	61	[47]
**Component (% dry wt basis)**												
Free sugars	5	[36]	ND	ND	ND	ND	ND	ND	ND	ND	ND	ND
Cellulose	36		8	[48]	12	[49]	28	[50]	24	[51]	16	[47]
Hemicelluloses	9		18		35		25		16		50 33% GM 17% AG	
Pectic polysaccharides	21		30		ND	ND	ND	ND	ND	ND	ND	ND
GalA (%)	80	[37]	52	[52]								
DE (%)	63		85									
DA (%)	6		6									
Mw (g mol^−1^)	400,000		12,000	[48]								
Total carbohydrates	71 *	ND	56 *	ND	47 *	ND	53 *	ND	40	[51]	66	[47]
Lignin	ND	ND	ND	ND	32	[49]	38	[50]	29		ND	ND
Melanoidins	ND	ND	ND	ND	ND	ND	ND	ND	5	[34]	16	[47]
Protein	9 **	[36]	17	[48]	ND	ND	8–11 **	[29,53]	19 **	[34,44,45,51]	5	
Lipids	0.8		ND	ND	ND	ND	1–3		2–5		13–15	[47,54]
Ash	ND		ND	ND	1	[55]	3–7	[29,55]	5–7		2	[56]
Caffeine	1	[36]	ND	ND	0.13	[42]	1	[29,53]	1	[57]	0.01–0.5	
Total phenolics (% *w*/*w* GAE)	0.3		ND	ND	0.2		1	[58]	2		1–2	

AG: arabinogalactans; DA: degree of acetylation; DE: degree of methyl esterification; GAE: gallic acid equivalents; GalA: Galacturonic acid; GM: galactomannans; Mw: Molecular weight; ND: Not determined. * Estimated from the sum of cellulose, hemicelluloses, pectic polysaccharides, and free sugars of each coffee by-product. ** Protein content may be overestimated due to the presence of other nitrogenous compounds in the same composition.

**Table 2 foods-10-00683-t002:** Functional additives for nonbiodegradable plastic formulations using crude coffee by-products.

By-Product	Coffee-Based Powder	Polymeric Matrix	Developed Materials and Main Properties	Ref
CH	CH powder	PP	Composites with poor interfacial adhesion between CH and the polymeric matrix	[71]
		PP (plus maleic anhydride grafted PP)	Composites with good interfacial adhesion	[72]
		HDPE (plus maleic anhydride grafted PE)	Composites with good interfacial adhesion	[73]
		PP (plus maleic anhydride grafted PP)	Composites with decreased susceptibility towards fire	[74]
		PP (plus maleic anhydride grafted PP)	Composites with decreased carbon footprint	[75]
	CH powder alkali treated	PP	Composites with improved mechanical and thermal performance	[76]
CS	CS powder	HDPE	Composites with poor interfacial adhesion between CS and the polymeric matrix	[77]
	CS powder alkali treated and esterified with palmitoyl chloride	HDPE	Composites with decreased water absorption	[77]
SCG	SCG powder	PP	Composites with poor interfacial adhesion between SCG and the polymeric matrix	[78]
	SCG powder treated by oil removal	PP (plus maleic anhydride grafted PP)	Composites with increase interfacial adhesion, compatibility, and water resistance	[79]
	SCG powder esterified with palmitoyl chloride	PP	Composites with better particle dispersion and decreased water uptake	[80]
	SCG powder alkali treated and bleached	PP (plus silane and styrene-ethylene-butene-styrene-graft-maleic anhydride)	Composites with improved interfacial adhesion and mechanical properties	[81]
	SCG powder treated by acid hydrolysis	PE	Antioxidant films with improved biocompatibility	[82]

CH: coffee husks; CS: coffee silverskin; HDPE: high-density polyethylene; PE: polyethylene; PP: polypropylene; SCG: spent coffee grounds.

**Table 3 foods-10-00683-t003:** Functional additives for biodegradable materials formulations using crude coffee by-products.

By-Product	Coffee-Based Powder	Polymeric Matrix	Developed Materials and Main Properties	Ref
CH	CH powder	PCL	Films with increased biodegradation rate	[86]
	CH powder treated with (3-glycidoxypropyl)trimethoxysilane	PBAT	Composites with increased hydrophobicity, stiffness, and reduced production cost	[87]
	Torrefied CH powder	PLA	Injection specimens with increased mechanical resistance and thermal stability	[88]
CH and CPm	CH and CPm powder	PHB	Composites with increased thermal stability and water absorption	[89]
CS	CS powder	PBAT and P(3HB-*co*-3HV)	Composites with antioxidant activity	[90]
	CS powder treated with (3-aminopropyl)triethoxysilane	PBAT and P(3HB-*co*-3HV)	Composites with antioxidant activity and increased interfacial adhesion	[91]
	CS powder	P(3HB-*co*-3HV)	Composites possessing an overall migration below the limit required for food packaging materials	[92]
	CS powder	Potato starch	Antioxidant and UV-protective films with increased elasticity, stretchability, and water resistance	[93]
SCG	SCG powder treated with tetraethyl orthosilicate	Maleic-anhydride-grafted PLA formulation	Homogeneous composites with increased water resistance and biodegradability	[94]
	Torrefied SCG powder	PBAT	Composites with increased hydrophobicity	[95]
	SCG powder	PVA (plus chitosan)	Homogeneous composites suitable for the adsorption of pharmaceuticals contaminants from water	[96]
		Corn starch	Films with increased tensile strength	[97]
		Cellulose	Photosensitive films	[98]
		Pectin	Films with increased water resistance	[99,100]

CH: coffee husks; CPm: coffee parchment; CS: coffee silverskin; P(3HB-*co*-3HV): poly(3-hydroxybutyrate-*co*-3-hydroxyvalerate); PBAT: polybutylene adipate terephthalate; PCL: polycaprolactone; PHB: polyhydroxybutyrate; PLA: polylactic acid; PVA: polyvinyl alcohol; SCG: spent coffee grounds.

**Table 4 foods-10-00683-t004:** Coffee by-products used as a source of extracts with film-forming ability or with functional properties for biodegradable materials formulations.

By-Product	Coffee-Based Molecules	Polymeric Matrix	Developed Materials and Main Properties	Ref
CP	Phenolic-rich extract	Chitosan	Films with increased water resistance, antioxidant, and antimicrobial properties	[108]
CM	Pectic polysaccharides	Pectic polysaccharides	Biodegradable films with rigidity and water insolubility	[52]
CPm	Phenolic-rich extract	Gellan gum	Films with improved antifungal properties	[42]
CH	Antioxidant and antibacterial aqueous extract	Corn starch	Antioxidant and antibacterial films with increased tensile strength and decreased water vapor and oxygen permeability	[58]
	Cellulose fibers	Corn starch	Films with increased stiffness	[58]
	Antioxidant and antibacterial aqueous extract	Corn starch/PLA	Antioxidant films with decreased oxygen permeability	[111]
	Cellulose nanocrystals	Corn starch/PLA	Films with increased tensile strength and decreased gas permeabilities	[111]
CS	Cellulose nanocrystals	PLA	Films with increased tensile strength and decreased gas permeability	[117]
SCG	Polysaccharide-rich extract	Carboxymethyl cellulose	Active brown films with increased light barrier, hydrophobicity, and tensile resistance	[118]
	Oil	PLA	Composites with increased toughness suitable for 3D-printing applications	[106]
	Fatty acids-rich extract	PLA (plus diatomite)	Films with increased interfacial adhesion and decreased oxygen permeability	[107]
	Phenolic-rich extract	PVA/cassava starch	Antioxidant, antimicrobial, and antibacterial films	[112]
		Cassava starch	Antioxidant, antimicrobial, and antibacterial films	[113]
	Polysaccharide-rich extract	Galactomannans	Heterogeneous films with light-brownish coloration	[64]
	Galactomannans-rich extract	Galactomannans	Heterogeneous and rigid films with light-brownish coloration	[65]

CH: coffee husks; CM: coffee mucilage; CP: coffee pulp; CPm: coffee parchment; CS: coffee silverskin; PLA: polylactic acid; PVA: polyvinyl alcohol; SCG: spent coffee grounds.

## Data Availability

Not applicable.

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
