# Peer review of "Coffee By-Products and Their Suitability for Developing Active Food Packaging Materials"

_foods, 2021, doi:10.3390/foods10030683_

Round 1
Reviewer 1 Report
The paper " Coffee byproducts and their suitability for developing active food packaging plastics" carries out a consistent review on the characteristics of the main byproducts of coffee processing industry, focusing on their potential applications in the field of packaging.
This paper accomplishes the scope of Foods, therefore I recommend minor modifications. The only recommendations are the following:
- Line 128. I suggest replacing “namely caffeine” with “such as caffeine and trigonelline’’;
- Line 209. I suggest considering further phenolic compounds found in SCG in the list, not only chlorogenic acids;
- Lines 375-390. The importance of antioxidant and antimicrobial activity of phenolics compounds and alkaloids for active food packaging purposes should be better highlighted;
- Conclusions and future perspective. Authors ‘considerations about how the mentioned applications could increase circularity and packaging sustainability should be added.
Author Response
Reviewer 1:
The paper " Coffee byproducts and their suitability for developing active food packaging plastics" carries out a consistent review on the characteristics of the main byproducts of coffee processing industry, focusing on their potential applications in the field of packaging.
This paper accomplishes the scope of Foods, therefore I recommend minor modifications. The only recommendations are the following:
- Line 128. I suggest replacing “namely caffeine” with “such as caffeine and trigonelline’’.
Line 209. I suggest considering further phenolic compounds found in SCG in the list, not only chlorogenic acids.
Answer: The authors thank the Reviewer’s comments. Although trigonelline contains nitrogen (C7H7NO2), the major contribution should be of caffeine (C8H10N4O2) due to its higher nitrogen content and amount in coffee byproducts. The sentence was reformulated for a better clarity. SCG phenolic composition includes mainly chlorogenic acids (85%), and a smaller fraction of caffeic acid (6%). This information was added to the manuscript.
- Lines 375-390. The importance of antioxidant and antimicrobial activity of phenolics compounds and alkaloids for active food packaging purposes should be better highlighted.
Answer: Plastics incorporated with phenolic-rich extracts have the potential to prevent food oxidation reactions when used as packaging due to the radical scavenging activity of coffee phenolic compounds. Also, coffee phenolic compounds and caffeine can confer food protection against microbial spoilage. This information has been added to highlight the importance of coffee phenolics and alkaloids in the development of active packaging with antioxidant and antimicrobial activity.
- Conclusions and future perspective. Authors‘ considerations about how the mentioned applications could increase circularity and packaging sustainability should be added.
Answer: The pointed aspects of circularity and packaging sustainability were emphasized in the conclusion section.
Reviewer 2 Report
The authors review the interesting and promising application of coffee by-products. The modern, ecologically oriented society attaches great importance to waste reduction, so it makes sense not to dispose of the by-products of coffee production and to bring them into the value chain. An added value of the coffee plant could increase social and economic prosperity in poorer coffee-growing regions and work against the decreasing coffee price, which is especially worthwhile in the current times of a global economic crisis.
The review is written in a comprehensive fashion with extremely helpful figures (excellent summary of the by products!) and tables, so that I have actually very few things to criticize apart from some suggestions:
- Another by-product that is available in high volume on the coffee farms are coffee leaves (e.g. from the regular cutting down of the trees). Do you see applications of this material for plastics as well?
- Abstract: some results and conclusions from the review could be added to the abstract. Currently, the abstract is rather uninformative. E.g. “the things are discussed”, but what was the result of the discussion?
- Figure 1: somehow, the percentages do not count up. The by-products are 100%, but where is the seed? Should this not count up to 29%?
- Section 2.1.1: if you have ever been on a coffee plantation, the rather poor quality of some of the by-products makes the application problematic. Sometimes, they are standing for days in the heat before processing and are highly spoilt by fementation and microorganisms. They probably also contain mycotoxins. For application, a more consistent quality control of the by products must be implemented.
- Line 198: why is this restricted to espresso?
- Line 215: typo: species
- Section 2.2.1: in the context of this section it must be warned that consumer deception may occur when such products are sold as biodegradable and “natural”. The same thing is currently happening with bamboo-polymers, which are more or less melamine-formaldehyde based but are sold as “bio-polymers”. They are also leaching high levels of monomers.
- Line 244: “of carbohydrates”
- Table 3: enlarge “Ref” column to avoid line break in “100”
- Conclusion: I agree. Hopefully, an upscaling will occur in the near future.
Author Response
Reviewer 2:
The authors review the interesting and promising application of coffee by-products. The modern, ecologically oriented society attaches great importance to waste reduction, so it makes sense not to dispose of the by-products of coffee production and to bring them into the value chain. An added value of the coffee plant could increase social and economic prosperity in poorer coffee-growing regions and work against the decreasing coffee price, which is especially worthwhile in the current times of a global economic crisis.
The review is written in a comprehensive fashion with extremely helpful figures (excellent summary of the by products!) and tables, so that I have actually very few things to criticize apart from some suggestions:
- Another by-product that is available in high volume on the coffee farms are coffee leaves (e.g. from the regular cutting down of the trees). Do you see applications of this material for plastics as well?
Answer: The authors thank the Reviewer’s comments and the perspective for the point of view of coffee plant growers. Coffee leaves seem a very promising source of bioactive compounds with antioxidant, anti-inflammatory, antihypertensive, antibacterial, and antifungal activities, with potential to be applied on active food packaging. A paragraph was included, and references were provided concerning the potential application of coffee leaves for this purpose.
- Abstract: some results and conclusions from the review could be added to the abstract. Currently, the abstract is rather uninformative. E.g. “the things are discussed”, but what was the result of the discussion?
Answer: The abstract was revised according to the main properties conferred to plastic materials by coffee-products-derived molecules.
- Figure 1: somehow, the percentages do not count up. The by-products are 100%, but where is the seed? Should this not count up to 29%?
Answer: The wet processing increases the mass of coffee byproducts by the addition of water. This effect is better explained in the revised text.
- Section 2.1.1: if you have ever been on a coffee plantation, the rather poor quality of some of the by-products makes the application problematic. Sometimes, they are standing for days in the heat before processing and are highly spoilt by fermentation and microorganisms. They probably also contain mycotoxins. For application, a more consistent quality control of the by products must be implemented.
Answer: Thank you very much for your pertinent comments. A sentence was included in section 2.1.1 and also in the conclusions.
- Line 198: why is this restricted to espresso?
Answer: Thank you for highlighting this error. The term ‘’espresso’’ was removed.
- Line 215: typo: species
Answer: The term was revised.
- Section 2.2.1: in the context of this section it must be warned that consumer deception may occur when such products are sold as biodegradable and “natural”. The same thing is currently happening with bamboo-polymers, which are more or less melamine-formaldehyde based but are sold as “bio-polymers”. They are also leaching high levels of monomers.
Answer: We greatly appreciate your comment. A sentence was included to highlight that this can be seen as a first step towards a full biobased and biodegradable formulation able to provide the same functionalities of the petroleum-based materials.
- Line 244: “of carbohydrates”
Answer: The term ‘’by’’ was replaced by ‘’of’’ as suggested.
- Table 3: enlarge “Ref” column to avoid line break in “100”
Answer: The ‘’Ref’’ column in Table 3 was enlarged to avoid any line break.
- Conclusion: I agree. Hopefully, an upscaling will occur in the near future.
Answer: Thank you very much for your comment.